# Acquisition and Spread of Antimicrobial Resistance: A *tet*(X) Case Study

**DOI:** 10.3390/ijms22083905

**Published:** 2021-04-09

**Authors:** Rustam Aminov

**Affiliations:** 1School of Medicine, Medical Sciences and Nutrition, University of Aberdeen, Aberdeen AB25 2ZD, UK; rustam.aminov@abdn.ac.uk; 2Institute of Fundamental Medicine and Biology, Kazan Federal University, Kazan 420008, Russia

**Keywords:** tetracyclines, next-generation tetracyclines, antimicrobial resistance, natural reservoirs, mobile genetic elements, horizontal gene transfer

## Abstract

Understanding the mechanisms leading to the rise and dissemination of antimicrobial resistance (AMR) is crucially important for the preservation of power of antimicrobials and controlling infectious diseases. Measures to monitor and detect AMR, however, have been significantly delayed and introduced much later after the beginning of industrial production and consumption of antimicrobials. However, monitoring and detection of AMR is largely focused on bacterial pathogens, thus missing multiple key events which take place before the emergence and spread of AMR among the pathogens. In this regard, careful analysis of AMR development towards recently introduced antimicrobials may serve as a valuable example for the better understanding of mechanisms driving AMR evolution. Here, the example of evolution of *tet*(X), which confers resistance to the next-generation tetracyclines, is summarised and discussed. Initial mechanisms of resistance to these antimicrobials among pathogens were mostly via chromosomal mutations leading to the overexpression of efflux pumps. High-level resistance was achieved only after the acquisition of flavin-dependent monooxygenase-encoding genes from the environmental microbiota. These genes confer resistance to all tetracyclines, including the next-generation tetracyclines, and thus were termed *tet*(X). IS*CR2* and IS*26,* as well as a variety of conjugative and mobilizable plasmids of different incompatibility groups, played an essential role in the acquisition of *tet*(X) genes from natural reservoirs and in further dissemination among bacterial commensals and pathogens. This process, which took place within the last decade, demonstrates how rapidly AMR evolution may progress, taking away some drugs of last resort from our arsenal.

## 1. Introduction

In 2009, I analysed the phylogeny of relatively few *tet*(X) genes (spelling of tetracycline-resistant determinants in this article follows the nomenclature by Levy and others [1]) discovered at that time, which belong to the A family of flavin-dependent monooxygenases (FMOs) which are widely distributed in a variety of natural microbiota [2]. This interest has been dictated by the fact that Tet X enzymes are capable of efficient inactivation of tetracyclines, including the representative of the third-generation tetracyclines, tigecycline [3,4]. This drug (the minocycline derivative 9-tert-butyl-glycylamido-minocycline, GAR-936), was approved by the FDA in 2005 and by the EMA in 2006. Tigecycline appeared to be very effective against a broad range of bacterial pathogens, including those resistant to the first- and second-generation tetracyclines [2]. Resistance levels among Gram-negative pathogens were considered to be low and, in general, caused by mutations leading to the overexpression of efflux pumps belonging to the resistance nodulation division (RND) family. The risk of horizontal transfer of this non-specific RND resistance machinery among bacterial pathogens (and among any other bacteria for that matter) is low because they belong to the core constituents of bacterial genomes, they are complex, and they have evolved to perform a variety of specific functions within the cell other than antimicrobial resistance. On the contrary, evolutionary trajectories of genes encoding FMOs are incongruent with the genes of core genomes, such as encoding translation machinery (16S rRNA gene, for example) and suggest past horizontal exchange and duplication events [2]. The current genetic context of *tet*(X) has been also analysed, showing a potential for horizontal gene transfer (HGT) [2]. It has been concluded that the above prerequisites could make FMOs/Tet Xs the most likely mechanisms for the emergence of a high-level resistance to third-generation tetracyclines among bacterial pathogens. Thus, a possible mobilisation of FMO-encoding genes from environmental reservoirs and their dissemination into pathogenic microbiota should be closely monitored and intervened if necessary.

The situation with *tet*(X) was reassessed in 2013 [5]. At that time, the first indication of movement of *tet*(X) into clinical settings was reported [6]. In a hospital in Sierra Leone, 21% of isolates from urinary tract infections were confirmed to be *tet*(X)-positive. The range of multidrug-resistant (MDR) Gram-negative pathogens included *Enterobacter cloacae*, *Comamonas testosteroni*, *Escherichia coli*, *Klebsiella pneumoniae*, *Delftia acidovorans*, *Enterobacter* sp., and other members of the Enterobacteriaceae and Pseudomonadaceae families. The authors also noted that tigecycline was not available locally, but 87% of pharmacies dispensed the first- and second-generation tetracyclines without prescription [6]. Thus, the selective pressure imposed by the older tetracyclines may have driven the rise and spread of *tet*(X), which confers resistance to the third-generation tetracycline. Moreover, the authors also indicated the presence of mobile genetic elements (MGEs) in some isolates, which could serve as vehicles for *tet*(X) transmission and dissemination.

Selection by older tetracyclines could have been responsible for the emergence of *tet*(X) in food animals as well. In fact, the presence of *tet*(X) in an animal pathogen, *Riemerella anatipestifer*, was reported even earlier, in 2010 [7]. This bacterium causes septicaemia *anserum exsudativa* [8], which results in major economic losses in duck production [9,10]. The *R. anatipestifer* strain was reported in 2010 [7]; however, it was actually isolated in 2005. The strain carries plasmid pRA0511, which encodes two chloramphenicol acetyltransferases, a multi-drug ABC transporter permease/ATPase, and Tet X. Analysis of seven *R. anatipestifer* genomic sequences available at that time revealed that *tet*(X) was present in three genomic sequences but was undetectable in the other four genomes [5]. A few years later the situation changed drastically: it has been found that 80.2% of *R. anatipestifer* isolates from ducks possess the *tet*(X) gene [11]. Thus, during this very short timeframe, enzymatic inactivation has become the main mechanism of tetracycline resistance in *R. anatipestifer*, well surpassing other mechanisms of resistance such as ribosomal protection and drug efflux.

In brief, in the 2013 *tet*(X) update [5], it has been concluded that the major selective force responsible for the dissemination of this gene, which encodes resistance to the newer tetracycline(s), is actually not the use of the newer, but older tetracyclines. The newer tetracycline(s) are considered to be drugs of last resort and their use is limited, usually confined to controlled clinical settings. On the contrary, the use of the older tetracyclines is much less regulated, and they are used in vast quantities, especially in agriculture [12]. This broader and more extensive selection base in agriculture may lead to the accelerated evolution toward resistance to newer antimicrobial(s). Moreover, there is a widespread horizontal gene exchange among different ecological compartments of the global microbiota [13], which may contribute to the dissemination of agriculturally selected antimicrobial resistance (AMR) genes to human microbiota. Agricultural use of antibiotics is considered to be one of the major contributors to the global AMR problem, leading to the rise of untreatable human infections, thus resulting in public health crises [14]. Thus, given our past failures to deal with antimicrobial resistance due to the lack of knowledge, currently we are in a better position to finally implement the measures necessary to preserve the power of newer antimicrobials [5].

## 2. Present Situation with *tet*(X)

Unfortunately, however, during the past two years there have been an explosive number of publications showing that the worst-case scenario with the extensive dissemination of *tet*(X) into pathogenic microbiota is actively taking place, especially in China [15,16,17,18]. Current epidemiological data indicate that the variants of *tet*(X), *tet*(X3)/(X4)/(X5/X6/X14), can be detected in various ecological compartments in China, including humans and animals (mainly pigs, chickens, and ducks) [11,15,16,17,18,19,20,21,22]. According to a recent epidemiological analysis [18], the highest occurrence of Tet X-producing isolates has been recorded in China—42, followed by Sierra Leone—5, USA—4, Hungary—3, the Americas—1, the Czech Republic—1, France—1, and Japan—1. Worryingly, these *tet*(X) variants confer higher resistance levels against tigecycline, they are located on MGEs, and they are beginning to emerge in known human pathogens such as *Acinetobacter* species (especially *A. baumannii*) and Enterobacteriaceae [23,24]. Even more troublesome is the ability of these Tet X variants to degrade other next-generation tetracyclines such as eravacycline [25] and omadacycline [26]. In 2018, the FDA approved the former drug for therapy of complex intra-abdominal infections, and the latter — for treatment of community-acquired bacterial pneumonia and acute skin and skin structure infections [27]. For example, *tet*(X5), which was detected in a clinical *A. baumannii* isolate from China in 2017, confers universal resistance to all classes of tetracyclines, including tetracycline, doxycycline, minocycline, tigecycline, eravacycline, and omadacycline [28]. The same resistance phenotypes were detected among *tet*(X3–X5)-carrying *Acinetobacter* species isolated from different ecological niches across China [19]. The *tet*(X4)-carrying *E. coli*, which was detected in several provinces in China, was found to be capable of degrading all tetracyclines, including tigecycline and the newly approved eravacycline [16]. This *tet*(X4) gene is harboured by the IncQ1 plasmid, which is highly transferable to a number of clinical and laboratory strains within the Enterobacteriaceae family. Moreover, in can be stably maintained in transconjugants under non-selective conditions, suggesting that the fitness cost of the plasmid carriage is not significant [16]. The rise of the easily transferable resistance to the recently approved latest generations of tetracyclines, which can be stably inherited by pathogenic bacteria, is of concern because this severely compromises the contemporary arsenal of drugs of last resort.

## 3. Diversity of FMO-Encoding and *tet*(X) Genes

The currently identified range of Tet X variants include Tet X–Tet X14, with a variable level of amino acid sequence identity, with the lowest similarity between Tet X and Tet X14 at 67% [29]. Presumably, this is a reflection of natural diversity of FMOs that have been acquired from different taxonomic entities and different ecological niches. It has been suggested, for example, that the pool of *tet*(X7-X13) genes is mainly associated with the gastrointestinal tract, while one of them, *tet*(X7), is also detectable in a clinical *Pseudomonas aeruginosa* isolate [30]. However, *tet*(X8–X13) are currently represented only by metagenomic sequences, and their association with disease-causing bacteria remains unclear.

The range of natural diversity of FMOs, to which Tet Xs belong, is extraordinary and includes a wide variety of enzymes involved in different oxygenation reactions [31]. These broad spectra are also reflected in the ability to destroy a range of antimicrobials, which presently include tetracyclines, rifampicins, sulfonamides, and β-lactams [32]. Besides tetracyclines, which are discussed here, these targets, for example, are represented by aromatic polyketides such as rifampicins [33,34,35]. Another class B FMO can oxidise the carbonyl moiety of the β-lactam ring, thus conferring resistance to imipenem [36]. The targets even include antimicrobials that are completely synthetic and have no pre-existing structural analogues in natural ecosystems such as sulphonamides. The well-known mechanism of sulphonamide resistance includes the *sul* genes encoding modified dihydropteroate synthases [37]. A newly emerging mechanism of resistance to sulphonamides involves a two-component FMO that can catabolise these antimicrobials, and the location of the corresponding genes in the vicinity of *traA* suggests a potential for HGT [38]. Similarly, a recently identified class D FMO is capable of degrading sulphonamides, and the corresponding gene, *sulX*, is associated with MGEs [39]. Thus, due to their abundance in the environmental microbiota and broad substrate capacity, FMOs are capable of inactivating a number of antimicrobials of different classes via oxygenation reactions [32]. Although there are substantial structural differences between the older and newer generations of tetracyclines, a variety of FMO/Tet X enzymes are capable of degrading all of them [19,28]. Compared to the inactivation of other classes of antimicrobials, however, FMOs that degrade tetracyclines appeared to be more reproductively successful in Darwinian terms and disseminated into a large group of Gram-negative bacteria, including pathogens.

## 4. Phylogeny of *tet*(X) Genes

Previous phylogenetic analyses suggested that the FMO-encoding genes from the group of environmental bacteria belonging to the Flavobacteriaceae family are most likely to be the origin of the *tet*(X) genes [18,19]. Indeed, the vast diversity of genes encoding tetracycline-inactivating monooxygenases resides within the species belonging to the Cytophaga, Fusobacterium, and Bacteroides (CFB) group bacteria (alternative naming—Bacteroidetes phylum, consisting of Bacteroidia, Flavobacteriia and Sphingobacteriia classes) (Figure 1 and Figure 2). It appeared to be that one species within the Flavobacteriia class, i.e., *R. anatipestifer,* is the most prominent bacterium in representing the majority of *tet*(X) diversity. The question that immediately arises then is whether this a natural diversity or a product of artificial selection? If it is a natural diversity, what could have been the driving force(s) behind this diversity? Bacteria may face a wide variety of xenobiotic compounds that have to be detoxified, and thus it is difficult to ascertain what the natural targets for FMOs have been before the start of industrial production of tetracyclines. Presently, no environmental factors that could have played this role could be admissibly propositioned. As mentioned before, this bacterium is a pathogen infecting mainly ducks and geese and causing a septicaemic disease, leading to significant losses in the industry. The recommended disease treatments are enrofloxacin in drinking water or subcutaneous/intramuscular injection of penicillin [40,41]. Thus, the disease treatment by these antimicrobials cannot be a direct selective force leading to the over-representation of *tet*(X) alleles in *R. anatipestifer* populations. The most probable selective force in this case could be tetracyclines which are extensively used for growth-promoting purposes, but not for disease treatment. This long-term selection by tetracyclines in duck farms may have resulted in the extensive diversification and enrichment of *R. anatipestifer* populations with a variety of *tet*(X) alleles that can currently be seen. There is a need to emphasise here that several Tet X clades consist entirely of this species (Figure 1 and Figure 2), from which *tet*(X) variants may potentially spread to other bacteria, including pathogenics.

In the E.U. countries, growth-promoting antibiotics were banned from January 1, 2006, when all these antibiotics, including tetracyclines, were deleted from the Community Register of authorised feed additives [42]. In the United States, tetracyclines were no longer allowed for growth promotion after January 1, 2017 [43]. Despite these regulatory measures, however, the use of antibiotics, including older tetracyclines such as chlortetracycline and oxytetracycline, in farm animals still remains a tenacious issue [44]. It is mainly associated with the current model of industrial production of inexpensive protein, which depends, to a great extent, on the use of antimicrobials. There are no specific data concerning tetracycline use in duck production in China, but it has been estimated [45] that tetracyclines (chlortetracycline and oxytetracycline) are the most used antibiotics in food animals there, including in broiler poultry (613,120 kg), swine (16,336,823 kg), and possibly others.

Another interesting aspect is the dissemination of *tet*(X) variants within the Tet X–X2 clade, which contains the *tet*(X) genes initially identified in *Bacteroides* species (Figure 1). The ancestral clade for it is represented by the corresponding protein in *R. anatipestifer*, with further entry into the other *R. anatipestifer* species and the representatives of the Bacteroidales order, consisting of *Bacteroides*, *Parabacteroides* and *Prevotella* species. These species are anaerobic gut bacteria, which raises a question as to why *tet*(X) is carried by these bacteria because the enzyme encoded requires oxygen and is presumably not functional under anaerobic conditions. 

Apparently, other flavobacteria may also serve as reservoirs of the *tet*(X) genes, from which they may be acquired by other, unrelated bacteria. For example, the ancestral clade for Tet X3, which includes *Acinetobacter* species, can be identified within a taxonomically unrelated flavobacterium, *Empedobacter brevis* (Figure 1). Within the clade, there are a large number of identical protein sequences. For example, the branch marked as “MULTISPECIES: tetracycline-inactivating monooxygenase Tet(X3) [Acinetobacter]” consists of 288 identical protein sequences detected in the genomes of different *Acinetobacter* species. This indicates the key role played by MGEs and HGT in the spread of *tet*(X3) among the *Acinetobacter* species.

The origin of some clades, however, could not be tracked back to the *R. anatipestifer* or another Flavobacteriia ancestry. Clade Tet X4, for example, comprises protein sequences that could be identified only within the enterobacteria and Gammaproteobacteria (Figure 1 and Figure 2). The MULTISPECIES branch consists of 409 identical proteins, 361 of which are identified in *E. coli* genomes. Other identical sequences can be detected in *Acinetobacter indicus*—4, *A. towneri*—2, *Acinetobacter* species—5, *Aeromonas caviae*—2, *Citrobacter braakii*—1, *C. freundii*—2, *Escherichia fergusonii*—1, *Klebsiella pneumoniae*—3, *K. quasipneumoniae*—1, various *Salmonella enterica* serovars—23, *Shigella flexneri*—2, and *S. sonnei*—2. Phylogenetic analysis suggests that the protein-encoding gene was acquired from an *E. coli* ancestor, with further dissemination into other *E. coli* strains, as well as to the representatives of enterobacteria and Gammaproteobacteria. The presence of the identical protein sequences in a diverse group of 409 bacteria within Gammaproteobacteria discussed above strongly supports the role of MGEs and HGT in dissemination of the *tet*(X4) gene.

Similarly, no involvement of any flavobacteria in the role of the *tet*(X) reservoir could be presumed for another clade, which consists solely of the Gammaproteobacteria representatives (Figure 2). In summary, these observations suggest the polyphyletic origin of *tet*(X) genes, which were recruited from natural reservoirs of FMO-encoding genes residing within different taxonomic entities. The currently identified sources are Flavobacteriia and Gammaproteobacteria, with the prominent role displayed by a representative of the former class, *R. anatipestifer.*

## 5. The Role of MGEs in Acquisition and Dissemination of *tet*(X) Genes

### 5.1. The Role of IS Elements in tet(X) Evolution

With the exception of the prototype *tet*(X/X2), genetic context surrounding other *tet*(X) genes is almost uniformly associated with IS*CR2* or IS*26*, occasionally in combination [15,16,17,18,19,23,24,28,47]. Sometimes, IS*CR2* is called IS*Vsa3* [48]; here, IS*CR2* will be used for the sake of consistency. IS*CR* (*CR* = common region) elements represent one of the most powerful gene-capturing systems, invented by bacteria, to sample and present the DNA acquired in a different cellular and genomic context [49]. Due to transposition via the rolling-circle replication mechanism, IS*CR*s, unlike the majority of IS elements, do not need two intact copies to transpose, and thus provide a very high rate of transposition compared to other IS elements. The process of gene sampling and presentation is random, but if its function is necessary in a new cellular environment, a gene becomes embedded in a new host. This is especially true for AMR genes that protect against the lethal action of antimicrobials. Indeed, IS*CR*s are linked to providing protective mechanisms against many antimicrobials of different classes [49].

The same authors also suggested that IS*CR* elements may have originated from aquatic bacteria [49]. This origin may be shared by a member of this family, IS*CR2*, as well. The wide distribution of IS*CR2* throughout multiple diverse aquatic genera, including *Vibrio*, *Shewanella*, *Pseudoalteromonas* and *Psychrobacter*, suggests its role in dissemination of AMR genes [50]. For example, one of the first florfenicol-resistant genes isolated, *floR*, was associated with IS*CR2*, and was detected on a plasmid from the fish pathogen *Pasteurella damsalae* subsp. *piscida*. This archetypal genetic context, with close association with IS*CR2* and *floR*, still persists, and can be frequently encountered in tigecycline-resistant bacteria, in combination with the recently acquired *tet*(X) variants [15,16,18,22,23,24,47].

Another genetic background associated with the *tet*(X) genes includes IS*26*, which is encountered less frequently than IS*CR2*, although sometimes these two ISs can be seen in combination, with IS*CR2* being encased by IS*26* [18,22,24]. IS*26* transposes via replicative mechanism and plays an important role in the reorganisation of plasmids that harbour genes, conferring resistance to a variety of antimicrobials [51]. IS*26* also preferentially transposes within plasmids over the chromosomes, thus contributing to HGT of AMR genes and the emergence of MDR phenotypes, especially in clinically significant bacteria. In the course of evolution, IS*26*s have been selected as a natural genetic engineering tool, allowing extensive plasmid restructuring and reassortment. The plasmid diversity generated serves as a starting material for the selection of plasmids that provide better protection of host cells against environmental challenges. Antimicrobial selection is one of these stern challenges, and the presence of IS*26-*generated diversity of plasmids with multiple drug-resistant determinants in clinical isolates [49] indicate a successful strategy implemented by bacteria to withstand the selective pressure of antimicrobials imposed by humans. More importantly, the role of IS*26* in restructuring and generation, for example, of novel multiple *tet*(X4)-carrying plasmid variants, was confirmed experimentally, during conjugation experiments under controlled laboratory conditions [24]. Thus, the combined effect of these two ISs, IS*CR2* and IS*26*, has been a major driving force that provided the rapid expansion of *tet*(X) into a variety of Gram-negative bacteria, some of which are known pathogens. 

### 5.2. The Role of Plasmids in tet(X) Dissemination

The next step in intra- and inter-species horizontal dissemination of AMR genes among bacteria involves plasmids, conjugative transposons, integrative conjugative elements, bacteriophages, and other MGEs capable of intercellular transmission of genetic information. One of the earliest indications of the role played by IS*CR2* and IS*26* in the acquisition and dissemination of *tet*(X) was obtained with plasmids, which were captured into *E. coli* recipients from soil microbiota by conjugation [52]. Interestingly, laboratory conjugation experiments with these plasmids showed a tendency for genetic rearrangements and chromosomal integration, presumably due to the presence of IS*CR2* and IS*26*. The authors concluded that *Acinetobacter* spp. were probably putative hosts for these plasmids in soil, which was a worrying sign because the closely related MDR *A. baumannii* was already identified as a major nosocomial infection and listed among ESKAPE pathogens requiring immediate attention [53,54,55].

Regrettably, however, the worst-case scenario has taken place during the past decade. Several *tet*(X) variants that are located on plasmids and confer high level resistance to the next-generation tetracyclines, tigecycline, eravacycline and omadacycline, can be found in human and animal isolates of *Acinetobacter* species, including *A. baumannii* [15,19,22,28]. Acquisition and dissemination of these *tet*(X) variants was mediated via IS*CR2* and IS*26* transposition into a variety of conjugative and mobilizable plasmids belonging to different incompatibility groups. Transfer of these *tet*(X)-bearing plasmids into *A. baumannii* strains circulating in the hospital environment may lead to the failure of therapies by drugs of last resort such as the next-generation tetracyclines. It should be noted here that IS*CR2* and IS*26* are highly spread among the clinical isolates of *Acinetobacter* spp., including *A. baumannii* [56], and these transposition mechanisms are crucial in supplying a rapid diversification of the acquired gene repertoire from other ecological compartments. Some of the genes that provide a better protection against antimicrobials could then be selected in, and inherited by, the survived host cells and assume the role of AMR genes. We still do not know exactly what the natural function/activity of FMOs, which are currently named as Tet Xs, is. In our practice, these genes have been found as conferring resistance towards several generations of tetracyclines; therefore, we call them *tet* genes, although their original function(s) in nature may be different.

Plasmid-mediated high-level resistance to the next-generation tetracyclines may become problematic for the corresponding treatment of Enterobacteriaceae infections as well. Enormous diversity of *tet*(X)-bearing plasmids of different incompatibility groups has been recently recovered just from a single slaughterhouse [24]. The plasmids isolated from 74 strains of *E. coli* and one strain of *Providencia rettgeri* were classified as ColE2-like, IncQ, IncX1, IncA/C2, IncFII, IncFIB, as well as hybrid plasmids with different replicons. Analogously to the case with *tet*(X)-bearing *Acinetobacter* species, this work emphasised the important role played by IS*CR2* and IS*26* in acquisition and dissemination of the *tet*(X) genes within the Enterobacteriaceae family as well. Moreover, the role of IS*26* in MDR plasmid reorganisation and generation of novel plasmid diversity was also experimentally confirmed [24].

Other studies have also reported a wide dissemination of *tet*(X)-bearing plasmids in Enterobacteriaceae, especially in *E. coli*, in a variety of environments, including food-producing animals and their environments, slaughterhouses, raw meat, hospitals, and in communities [15,16,17,24,47,57]. Interestingly, resistance to the next-generation tetracyclines in *E. coli* plasmids is almost exclusively encoded by the *tet*(X4) gene, while a broader diversity of *tet*(X) genes is carried by plasmids within *Acinetobacter* species.

### 5.3. The Role of Conjugative Transposons and Integrative Conjugative Elements (ICEs)

The initially identified Tet X1 and Tet X2-encoding sequences were located on a CTnDOT transposon from a human gut commensal bacterium *Bacteroides thetaiotaomicron*, which may also be widely distributed in a variety of other intestinal *Bacteroides* species [58]). The *tet*(X) gene in an environmental *Sphingobacterium* sp. strain was also found on a transposon-like element, Tn*6031,* which is very similar to CTnDOT [59].

A recent large-scale surveillance of clinical *tet*(X)-carrying Gram-negative bacterial isolates showed that the largest proportion of these genes can be seen among the representatives of the Flavobacteriaceae family compared to Enterobacteriaceae or *Acinetobacter* species [18]. In Flavobacteriaceae, the genetic context of *tet*(X) was usually associated with the chromosomally located ICEs. Additionally, unlike the plasmid-located *tet*(X) genes, there were hardly any traces of association with IS*CR2* and IS*26*, except for two strains of *Chryseobacterium bernardetii*, where partially deleted IS*CR2*s were detectable. Although the Flavobacteriaceae strains reported were of clinical origin, representatives of this family are usually environmental bacteria and inhabit various aquatic and terrestrial ecosystems. Most likely, these isolates were also of environmental origin, causing opportunistic infections in immunocompromised patients. In general, however, the role of conjugative transposons and ICEs in the dissemination of *tet*(X) seems less prominent compared to the plasmid-mediated HGT.

## 6. Emergence and Rise of AMR: From Chromosomal Mutations to Acquired Mobile Resistance

During the last decade, we had a unique opportunity to monitor the real-time evolution and dissemination of an AMR gene, *tet*(X). All previous examples of this kind have been severely limited, mainly due to the delayed detection, often only after a *post factum* emergence in pathogenic microbiota. Potential reservoirs, especially environmental, and the ways of acquisition of AMR genes from these reservoirs are difficult to reconstruct, if no extensive monitoring efforts are implemented. The long-term and widespread use of antimicrobials make the monitoring programs even more difficult because of extensive coverage necessary and the lack of reference points in most cases. Only in a very limited number of cases it is possible to reveal the original reservoir and putative transfer event resulting in the acquisition of an AMR gene by bacterial pathogens. This attainment could be, for example, illustrated by the emergence and dissemination of the mobile *qnr* genes, which confer resistance to quinolones. This has been initially described in a clinical *K. pneumoniae* strain isolated in 1994 [60]. Further analysis of *qnrA*-like genes established that an aquatic bacterium, *Shewanella algae*, is the natural reservoir for the *qnrA* genes [61]. Within the currently recognised seven families of *qnr* genes, the origin of some of them could be tracked to a variety of taxonomically and ecologically distant bacteria such as *C. freundii* (*qnrB*), Vibrionaceae (*qnrC*), *Enterobacter* spp. (*qnrE*), and *Vibrio splendidus* (*qnrS*) [62,63]. The most probable functional role of these pentapeptide repeat proteins in nature is the involvement in some steps of DNA metabolism involving stress responses, which also happened to be useful in the protection of DNA replication against quinolones. Initially, however, resistance to quinolones was due to chromosomal mutations, leading to target modification or by either decreased uptake or increased efflux. Only later were these resistance mechanisms enhanced by the acquisition and wide dissemination of transferrable quinolone-resistant genes mediated by MGEs. Initially, these mechanisms remained obscure, but from 1998 onward they spread in epidemic proportions [62]. This enabled breaking of the constraints imposed by chromosomal mutations, which restricted the spread of AMR to vertical inheritance and clonal dissemination. Instead, a widespread dissemination of AMR became possible that overcomes taxonomic and ecological barriers. 

Similarly, the initial mechanisms of tigecycline resistance were also due to chromosomal mutations that led to the overexpression of efflux pumps. In clinical isolates of *A. baumannii*, for example, the initial reduced susceptibility towards the drug was due to the elevated expression of the RND family of efflux pumps, AdeABC and AdeIJK [64,65]. In *E. cloacae* and *E. coli* as well, reduced tigecycline susceptibility was due to the elevated expression of another efflux pump in the RND family, AcrAB [66,67]. In general, these efflux mechanisms are non-specific, and their overexpression is due to chromosomal mutations affecting the regulatory network [64]. The efflux systems, which are chromosomally located, consist of several genes, and they are tightly integrated into the cellular regulatory network, with limited chances for acquisition and dissemination via MGEs. Even in the original hosts, the non-specific drug efflux only offers marginal protection. However, these mechanisms provide intermediate protection and survival, before the arrival of genes that are capable of delivering high-level drug resistance. Generation of such a potential gene pool and its exposure to other cellular contexts is a continuous process driven by MGEs, which results in extensive HGT within and among different ecological compartments [13]. Some genes in a new cellular environment thus could occasionally confer resistance to antimicrobials and then are selected as AMR genes. This scenario has been possibly instigated in the case of FMOs, some of which happened to have enzymatic activities capable of destructing certain antimicrobials including the next-generation tetracyclines [32]. This chain of events leading to the rise and dissemination of *tet*(X) is depicted in Figure 3.

An initial step in the process of acquisition of mobile resistance towards all tetracyclines could have been the in situ selection and enrichment of natural reservoirs of *tet*(X) in order to reach the critical threshold density, which is crucial for MGEs to operate via conjugation mechanisms and provide successful HGT. This enrichment, especially in flavobacteria and Gammaproteobacteria, probably resulted from the extensive selection by older tetracyclines, and most likely in agricultural settings where they are mainly used, both in proportional and absolute terms. Analysis of the *tet*(X) diversity (Figure 1 and Figure 2) demonstrated that it largely resides within *R. anatipestifer.* A recent retrospective screening of 212 *R. anatipestifer* isolates from 58 large-scale duck farms in different regions of China demonstrated that 90.6% of them are resistant to tetracycline [11]. Interestingly, 80.2% of these isolates possessed the *tet*(X) gene, while the next most frequent tetracycline-resistant gene, *tet*(A), was encountered only in 20.8% of isolates. Moreover, if isolates that carry *tet*(X) were counted in combination with other tetracycline-resistant genes, the total *tet*(X)-carriage rate by this bacterium reaches 99.6%. Moreover, *tet*(X) could be transferred to a tetracycline-susceptible *R. anatipestifer* strain by conjugation, as well as by natural transformation. These prerequisites make *R. anatipestifer* the most prolific reservoir of mobile *tet*(X) genes. The corresponding reservoirs in Gammaproteobacteria are less understood but presumably they are similar, although apparently not as abundant. However, some representatives of these bacteria were very successful in the role of pathogens, and they possess highly sophisticated natural genetic engineering tools. These provisions make them formidable opponents, especially in light of the acquisition of resistance to drugs of last resort. Some of the ESKAPE bacteria such as *A. baumannii, K. pneumoniae* and *P. aeruginosa* have already acquired the *tet*(X) genes that confer resistance to the recently approved next-generation tetracyclines. This may severely limit our options to control these pathogens. 

The widespread presence of different variants of Tex X in *R. anatipestifer* (Figure 1 and Figure 2) suggests that there is a positive selection for these tetracycline-resistant genes. Although why have other *tet* genes not been selected under the same selective pressure? Do these FMOs confer some sort of selective advantage beyond tetracycline resistance? It has been shown, for example, that knocking out the tigecycline-resistant genes in *R. anatipestifer* results in a decrease in bacterial metabolic activity [68]. In general, however, the proportion of the *tet*(X) genes is quite small within the known and computed diversity of all tetracycline-resistant genes [69]. Additionally, there is no indication of *tet*(X) entry into Gram-positive bacteria. Is it because the natural pool of FMOs in these bacteria is rather limited? Or is the bottleneck in the MGEs of Gram-positive bacteria? These questions merit further investigation for better understanding of AMR evolution.

## 7. Conclusions

Witnessing the rise and dissemination of high-level resistance towards the next-generation tetracyclines within the last decade gave us a unique opportunity to understand some key aspects of AMR evolution. Initially, non-specific AMR mechanisms are involved that are chromosomally encoded, non-transferable, and can provide only low-level resistance. Continuous reshuffling of genetic material among different ecological compartments via MGEs and HGT may bring a chance of encountering gene(s) that are good at protecting the host against antimicrobials (high-level resistance) and easily transferable. Then, these selected genes can be transferred to other microbiotas, including pathogenic. Can this chain of events be interrupted? This is possible, but it requires careful rethinking of and alterations in the current supply chains [44]. The complexity of the problem requires concerted efforts at various levels, including research, education, governmental and regulatory agencies, health professionals, veterinarians, agricultural specialists, and other stakeholders.

## Figures and Tables

**Figure 1 ijms-22-03905-f001:**
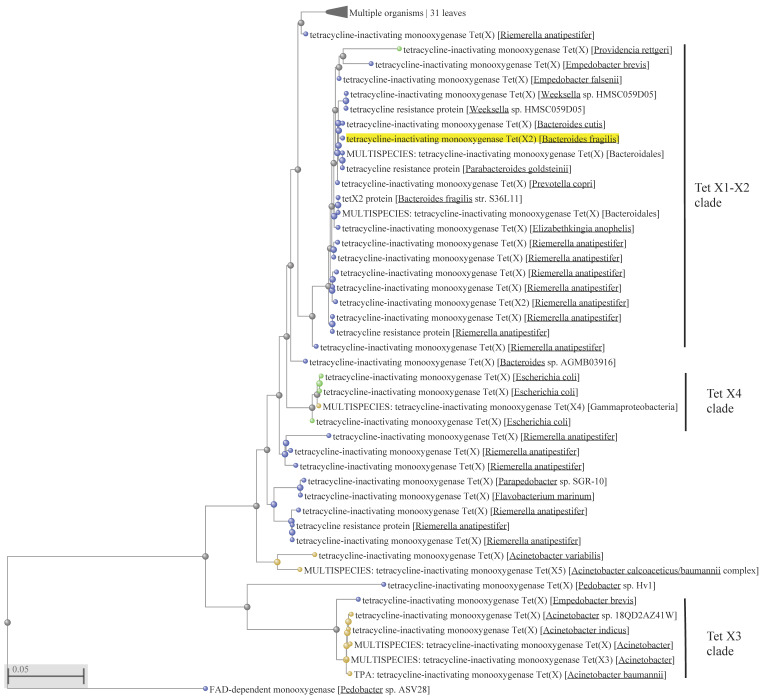
Phylogenetic reconstruction of flavin-dependent monooxygenase (FMO)/Tet X evolution produced with fast minimum evolution algorithm [46] using FMO and Tet X amino acid sequences predicted from gene sequences. The dataset of 76 non-redundant amino acid sequence was retrieved from GenBank using the Tet X2 sequence from *Bacteroides fragilis* (GenBank accession number WP_063856436) as a query (highlighted in yellow). Uncultured/environmental sample sequences were excluded from the analysis. Sequences were aligned using Constraint-based Multiple Alignment Tool (Cobalt) at the NCBI site (https://www.ncbi.nlm.nih.gov/tools/cobalt/cobalt.cgi, accessed on 8 March 2021). Amino acid positions from 11 to 388 were used to construct the tree. FAD-dependent monooxygenase from *Pedobacter* sp. ASV28 (GenBank accession number WP_199119858.1) was used to root the tree. MULTISPECIES indicates that identical amino acid sequences were present in multiple species. The scale bar is in fixed amino acid substitutions per sequence position. Nodes and branches of the CFB group bacteria (Bacteroidetes phylum) are in blue, Gammaproteobacteria in yellow, Enterobacteria in green, and unspecified bacteria in grey.

**Figure 2 ijms-22-03905-f002:**
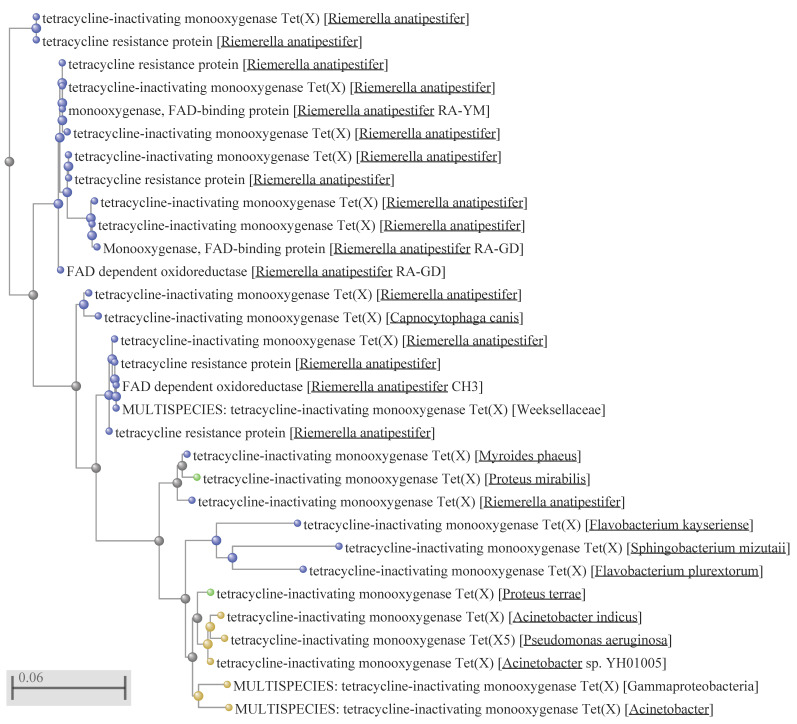
Expansion of the branch with 31 organisms in Figure 1. All corresponding designations are the same as in Figure 1.

**Figure 3 ijms-22-03905-f003:**
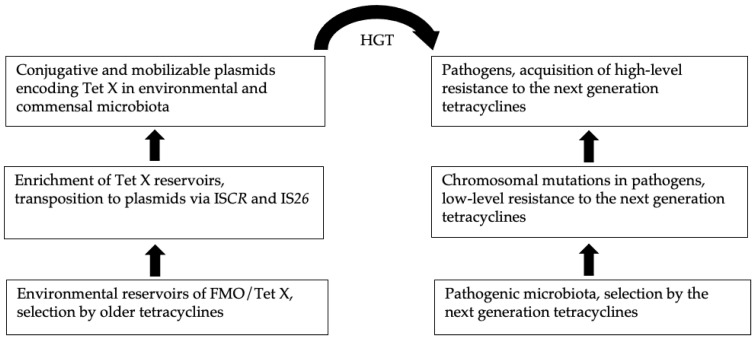
Schematic representation of *tet*(X) evolution.

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
