# Peer review of "Acquisition and Spread of Antimicrobial Resistance: A *tet*(X) Case Study"

_ijms, 2021, doi:10.3390/ijms22083905_

Round 1
Reviewer 1 Report
This a very good review about tet(x) acquisition and spread of antimicrobial resistance.
I have several questions:
- "P3 Current epidemiological data indicate that the variants of tet(X), tet(X3)/(X4)/(X5/X6/X14), can be detected in various ecological compartments in China, including humans and animals." Would you list the name of animal ? And for the readers, would you highlight if the tet(X) has been spread outside China ? And in other animal (not duck) , can tet(X) be found in Riemerella anatipestifer or other bacteria (E. coli ) ?
- "p4: The most probable selective force in this case could be tetracyclines that are extensively used for growth-promoting purposes, not for disease treatment. Similarly, P2 the use of older tetracyclines is much less regulated, and they are used in vast quantities, especially in agriculture (Koike et al., 2017) What is the common tetracyclines use in growth promoting or in the duck ? Old tetracycline or structure similar to new tetracycline e.g tigecycline/omandacycline ? Would you more specify the name of tetracycline use in the food animal ?
Author Response
"P3 Current epidemiological data indicate that the variants of tet(X), tet(X3)/(X4)/(X5/X6/X14), can be detected in various ecological compartments in China, including humans and animals." Would you list the name of animal ? And for the readers, would you highlight if the tet(X) has been spread outside China ? And in other animal (not duck) , can tet(X) be found in Riemerella anatipestifer or other bacteria (E. coli ) ?
Response:
The sentence is modified to include the animal names and references: “Current epidemiological data indicate that the variants of tet(X), tet(X3)/(X4)/(X5/X6/X14), can be detected in various ecological compartments in China, including humans and animals (mainly pigs, chickens, and ducks)".
This sentence was incorporated into the text: “According to a recent epidemiological analysis (Zhang et al., 2020), the highest occurrence of Tet X-producing isolates is recorded in China – 42, followed by Sierra Leone – 5, USA – 4, Hungary – 3, Americas – 1, Czech Republic – 1, France – 1, and Japan – 1.”
And the last point: yes, it is clearly stated in the manuscript, that tet(X) can be encountered in different bacteria such as Bacteroidetes, Proteobacteria, and Flavobacteria. These bacteria present in a variety of environments, including food-producing animals such as pigs, chickens and ducks and their environments, slaughterhouses, raw meat, hospitals, and in communities.
p4: The most probable selective force in this case could be tetracyclines that are extensively used for growth-promoting purposes, not for disease treatment. Similarly, P2 the use of older tetracyclines is much less regulated, and they are used in vast quantities, especially in agriculture (Koike et al., 2017) What is the common tetracyclines use in growth promoting or in the duck ? Old tetracycline or structure similar to new tetracycline e.g tigecycline/omandacycline ? Would you more specify the name of tetracycline use in the food animal ?
To clarify these questions, the next sentences were incorporated into the manuscript:
“In the EU countries, the growth-promoting antibiotics were banned from January 1, 2006, when all these antibiotics, including tetracyclines, were deleted from the Community Register of authorized feed additives (Castanon, 2007). In USA, tetracyclines are no longer allowed for growth promotion after 1 January 2017 (Granados-Chinchilla and Rodríguez, 2017). Despite these regulatory measures, however, the use of antibiotics, including older tetracyclines such as chlortetracycline and oxytetracycline, in farm animals still remains a tenacious issue (Kirchhelle, 2018). It is mainly associated with the current model of industrial production of inexpensive protein, which depends, to a great extent, on the use of antimicrobials.”
“Although there are substantial structural differences between the older and newer generations of tetracyclines, a variety of FMO/Tet X enzymes are capable of degrading all of them (Wang et al., 2019; Chen et al., 2020).”
“There are no specific data concerning tetracycline use in duck production in China, but it has been estimated (Krishnasamy et al., 2015) that tetracyclines (chlortetracycline and oxytetracycline) are the most used antibiotics in food animals, including broiler poultry (613,120 kg), swine (16,336,823 kg), and possibly others.
The corresponding new references were also incorporated:
Castanon JI. 2007. History of the use of antibiotic as growth promoters in European poultry feeds. Poult Sci 86:2466–2471. 10.3382/ps.2007-00249.
Granados-Chinchilla F, Rodríguez C. Tetracyclines in Food and Feedingstuffs: From Regulation to Analytical Methods, Bacterial Resistance, and Environmental and Health Implications. J Anal Methods Chem. 2017;2017:1315497. doi:10.1155/2017/1315497
Kirchhelle, C. Pharming animals: a global history of antibiotics in food production (1935–2017). Palgrave Commun 4, 96 (2018). https://doi.org/10.1057/s41599-018-0152-2.
Krishnasamy, V., Otte, J. & Silbergeld, E. Antimicrobial use in Chinese swine and broiler poultry production. Antimicrob Resist Infect Control 4, 17 (2015). https://doi.org/10.1186/s13756-015-0050-y
All these changes in the manuscript are marked in yellow.
Reviewer 2 Report
In this review, Dr. Aminov describes that agricultural use of older tetracyclines is considered to be one of the major contributors to the global antimicrobial resistance (AMR) problem leading to the rise of untreatable human infections thus resulting in public health crisis. Generally speaking, the review is well written and should contribute to the field of AMR studies. All references were appropriately cited. Before this review can be accepted for publication in IJMS, there are some minor points that needed to be checked as follows.
- In Figure 1 & 2, the numbers 0.05 and 0.06 (probably scale bar) needs to be defined in the figure legend.
- In Figure 1 legend, it stated that “Nodes and branches of the CFB group bacteria (Bacteroidetes phylum) are in blue, Gammaproteobacteria – in yellow, Enterobacteria – in green, and unspecified bacteria – in grey”. However, no color can be seen in the phylogenetic tree presented.
Author Response
- In Figure 1 & 2, the numbers 0.05 and 0.06 (probably scale bar) needs to be defined in the figure legend.
Response:
These numbers are actually defined in the figure legend: "The scale bar is in fixed amino acid substitutions per sequence position."
- In Figure 1 legend, it stated that “Nodes and branches of the CFB group bacteria (Bacteroidetes phylum) are in blue, Gammaproteobacteria – in yellow, Enterobacteria – in green, and unspecified bacteria – in grey”. However, no color can be seen in the phylogenetic tree presented.
Response:
Unfortunately, in the pdf version, the colour coding disappeared during conversion to pdf. It is present, however, in the original .docx file. Please refer to this version for colour coding.